# Ecological interactions between marine RNA viruses and planktonic copepods
Junya Hirai [1] ✉, Seiji Katakura[2], Hiromi Kasai[3] & Satoshi Nagai[4]

The interactions between zooplankton and viruses, which have been overlooked despite their crucial roles in marine ecosystems, are investigated in the copepod *Pseudocalanus newmani*. Copepod transcriptome data reveal four novel RNA viruses and weekly zooplankton samplings detect all viruses with different prevalence peaks during low-abundance periods of *P. newmani*. In addition to water temperature and food quality, our results suggest that marine virus is one of the factors controlling copepod population dynamics. Gene expression analysis indicates possible increased viral replication and decreased copepod movement in *P. newmani* with the Picorna-like virus, which is closely related to phytoplankton viruses, and metabarcoding diet analysis detects diatoms as *P. newmani*'s major prey. Viral-like particles are observed in the gut contents of copepods during the high prevalence of this virus, suggesting infected copepod prey may affect copepod physiology. These results show that investigating zooplankton–virus interactions can provide a better understanding of marine ecosystems.

Since the discovery of viruses with high abundance in the oceans[1], marine viruses have been recognized as key players in marine food webs and biogeochemical cycles[2]. High bacterial mortality rates of up to approximately 40% have been estimated[3], and viral infections in phytoplankton induce primary production loss[4]. Additionally, pathogenic viruses have caused devastating damage to the aquaculture of economically important organisms, including fish and shrimp[5,6]. However, the ecological roles of marine zooplankton-associated viruses are poorly understood, despite the importance of zooplankton as a link between lower and higher trophic levels in marine food webs. Planktonic copepods are particularly dominant among marine zooplankton, and the population dynamics of key copepod species are largely linked to the recruitment success of commercially important fish[7]. Top-down predation is a major controlling factor of population dynamics in planktonic copepods that accounts for 2/3–3/4 of the total mortality[8]. However, up to 1/3 of the non-predatory mortality in copepods has not been completely explained, and high copepod carcass proportions have been reported in various marine environments[9]. Further investigations on virus–copepod interactions should be conducted to provide a better understanding of the ecological importance of marine viruses in marine ecosystems.

Studies on marine viruses have mostly focused on microbial communities and commercially important organisms. Additionally, planktonic copepods have been reported to function as vectors for phytoplankton, shellfish, and large crustacean viruses[10–12]. In terrestrial ecosystems, some plant viruses can replicate and cause immune responses in insect vector species[13]. However, few studies focused on the influences of viruses on

vectors of marine planktonic copepods, and no negative effects on the fecundity and survival of copepods incubated in seawater with concentrated virus-like particles have been observed[14]. The early study on viruses infecting marine planktonic copepods demonstrated a high prevalence and copy number of ssDNA circo-like viruses in two copepod species, with temporal variability in viral infections[15]. dsDNA *Iridoviridae* viruses from estuarine copepods have a possible impact on copepod swimming activity, although detailed investigations have not been undertaken[16]. Additionally, recent comprehensive genetic analyses suggest that RNA viruses interact with zooplankton, especially copepods, in pelagic realms, which act as a reservoir of marine viruses and influence virus evolution[17–19]. The potential importance of crustacean zooplankton viruses in the global ocean has been reviewed[20]; however, the ecological impacts of viruses associated with zooplankton, including copepods, remain poorly understood. The population dynamics of *Daphnia* species and viruses were investigated in freshwater ecosystems[21]; thus, detailed copepod-virus ecological dynamics should be investigated by focusing on the key copepods to further understand the ecological roles of marine viruses.

Given the ecological importance of copepods and viruses, we hypothesized that marine viruses are linked to the population dynamics and physiology of planktonic copepods and play a pivotal role in the marine ecosystem. To shed light on these hidden copepod-virus interactions, extensive samplings for zooplankton and environmental conditions were performed in the coastal area of the Okhotsk Sea (Supplementary Fig. 1). We focused on *Pseudocalanus newmani*, a dominant copepod with large

[1]Atmosphere and Ocean Research Institute, The University of Tokyo, Kashiwa, Japan. [2]City of Mombetsu, Mombetsu, Japan. [3]Fisheries Resources Institute, Japan Fisheries Research and Education Agency, Kushiro, Japan. [4]Fisheries Technology Institute, Japan Fisheries Research and Education Agency, Yokohama, Japan. ✉e-mail: hirai@aori.u-tokyo.ac.jp

population dynamics in the coastal waters of the study area. Being a crucial source of food for fish larvae[22], *P. newmani* represents an ideal species for investigating the role of copepod viruses in marine ecosystems. First, we detected the major viruses associated with *P. newmani* in the transcriptome data. Second, we investigated the population dynamics of *P. newmani* and major viruses using weekly zooplankton samples to determine the impact of viruses on copepod abundance. Third, we revealed the physiological changes in *P. newmani* based on gene expression patterns under high viral loads. Additionally, we investigated and compared other factors, including environmental conditions, genetic populations, predators, and eukaryotic parasites, with the population dynamics of *P. newmani* to clarify the ecological roles of copepod-associated viruses.

## Results

### Detecting the major viruses

Viral sequences were detected from rRNA-depleted transcriptome data of *P. newmani* after incubation for gut depuration. More than 122 million quality-filtered reads were assembled into 856,835 contigs. Among the seven viral contigs, four were selected as the major viruses (Supplementary Table 1): PSNE-Narna (3221 bp), a Narna-like virus; PSNE-Pico1 (9182 bp) and PSNE-Pico2 (7676 bp), two Picorna-like viruses; and PSNE-Toga (11,039 bp), a Toga-like virus. All major viruses were newly reported RNA

viruses that exhibited < 60% identity to the amino acid sequences of known viruses. Based on the mapped reads, PSNE-Toga was the most dominant contig, followed by PSNE-Narna and PSNE-Pico 1 and 2 (Supplementary Table 1). Phylogenetic analysis based on the RdRp gene in *Picornavirales* classified PSNE-Pico1 as a member of the family *Marnaviridae* (Fig. 1a). PSNE-Pico1 was phylogenetically related to *Bacillarnavirus* and presented the same genome structure of two open reading frames (ORFs) (Fig. 1b)[23]. PSNE-Pico2 was not classified into any known viral family in *Picornavirales* and formed a new genetic lineage with other invertebrate viruses (Fig. 1c)[24]. The virus most closely related to PSNE-Pico2 was detected in the calanoid copepod *Cosmocalanus darwinii*[17]. PSNE-Pico2 contains one ORF, and its genome structure is the same as that of other invertebrate viruses in this genetic lineage. PSNE-Narna was distinct from the known group *Narnavirus* and phylogenetically similar to Narna-like viruses obtained from plant and invertebrate transcriptome data[24,25]. Additionally, the genome structure of one ORF containing the RdRp gene was shared among these viruses (Fig. 1d). PSNE-Toga, which has one long ORF, was not classified into any known groups of *Togaviridae* that present two ORFs for non-structural and structural polyproteins. The genome structure of PSNE-Toga was shared with that of other Toga-like viruses obtained from house dust mites and flatworms despite a large genetic distance between PSNE-Toga and other Toga-like viruses obtained from invertebrates (Fig. 1e)[26,27].

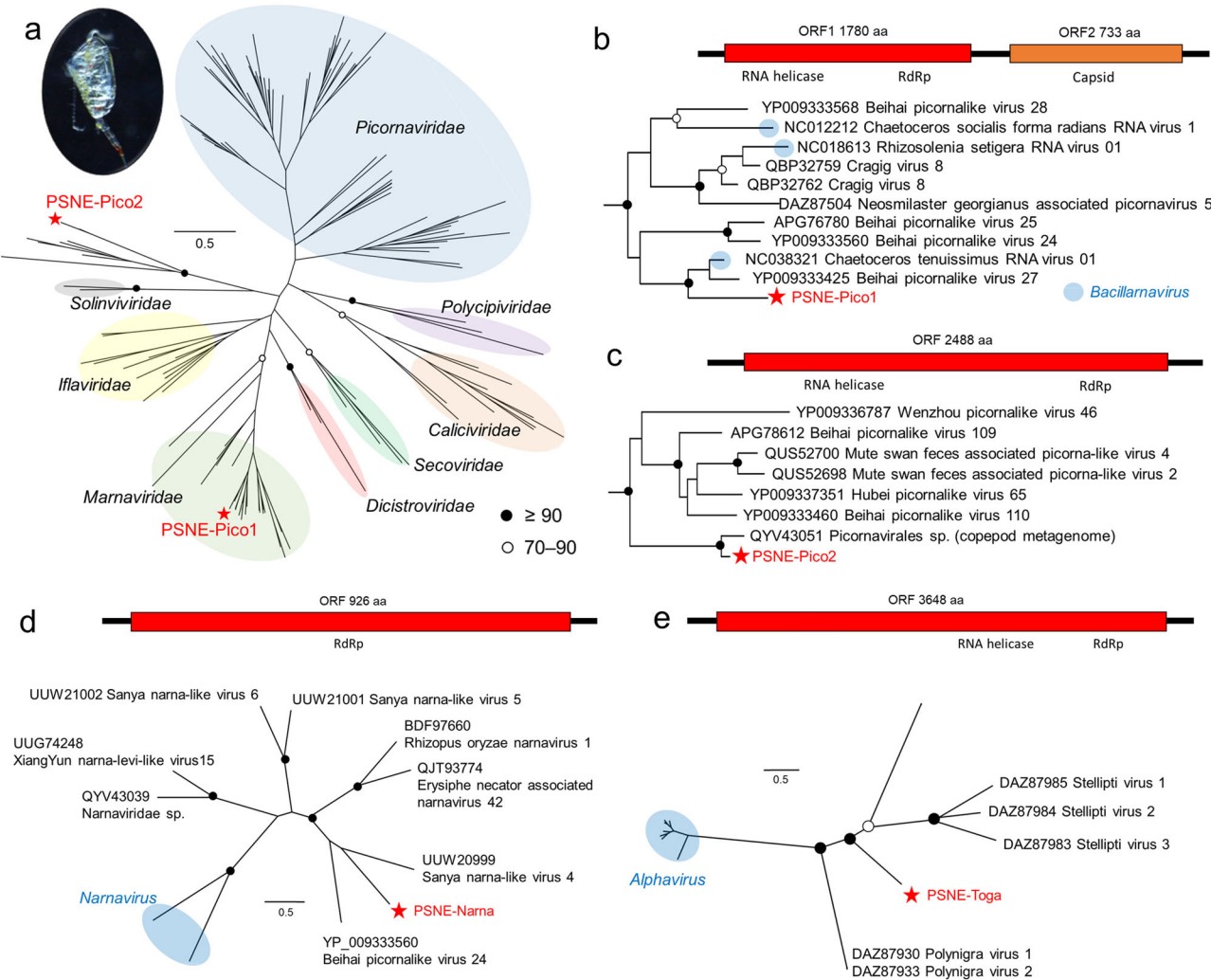

**Fig. 1 | Phylogenetic analyses of the detected viruses. a** Maximum likelihood tree for phylogenetic relationships among the members of *Picornavirales* and their relatives using RdRp amino acid sequences observed in the copepod *Pseudocalanus newmani*. Detailed phylogenetic relationships and genome structure of (**b**) PSNE- Pico 1, (**c**) PSNE-Pico 2, (**d**) PSNE-Narna, and (**e**) PSNE-Toga. Bootstrap values from the maximum likelihood analyses are indicated at ≥ 70%. The scale bar indicates the genetic distances. The best-fitted substitution model was LG + I + G4 + F (*Picornavirales* and PSNE-Toga) and PMB + I + G4 + F (PSNE-Narna).

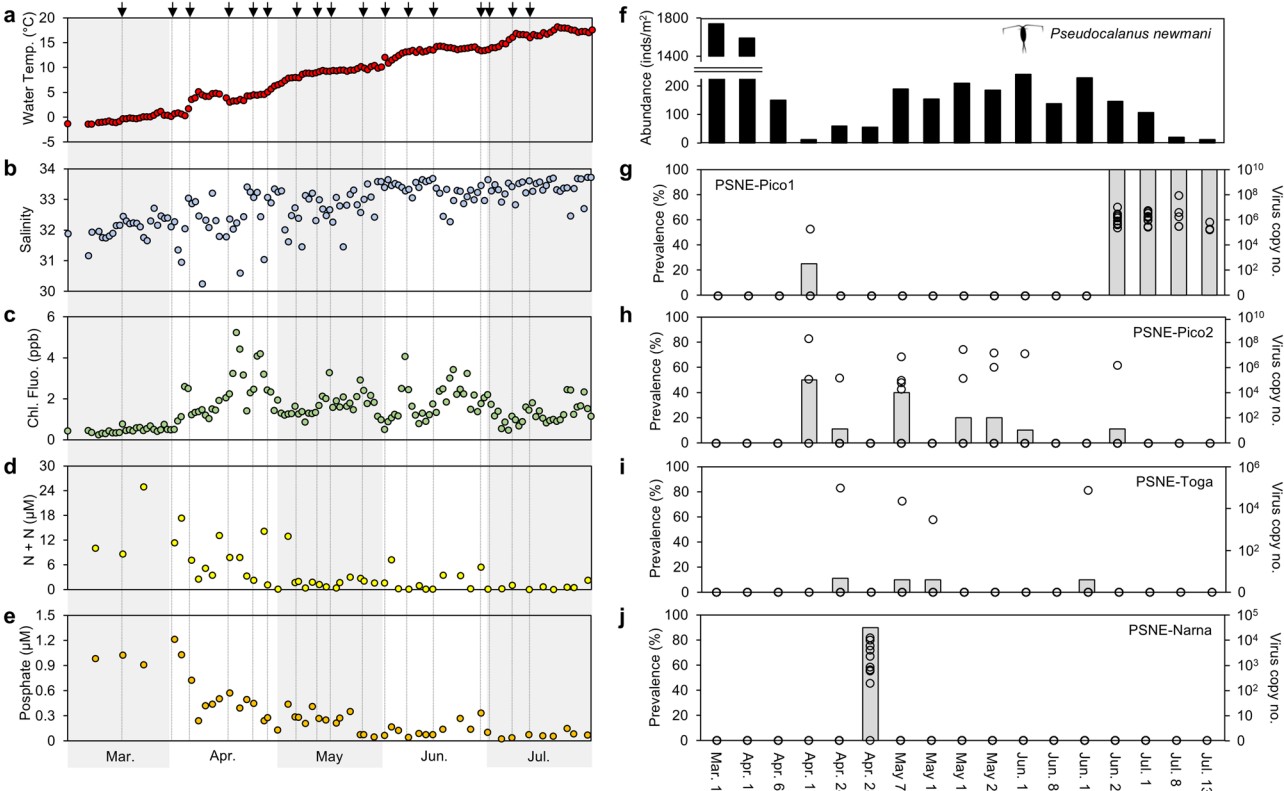

**Fig. 2 | Population dynamics of copepods and viruses during the sampling periods with different environmental conditions.** Environmental variables include (**a**) water temperature, (**b**) salinity, (**c**) chlorophyll fluorescence, (**d**) nitrate + nitrite (N + N), and (**e**) phosphate from March to July 2020. Black arrows indicate the

sampling days of the zooplankton. In each sampling day of zooplankton, (**f**) abundances of the adult female *Pseudocalanus newmani* are compared with (**g–j**) the copy numbers and prevalence of major viruses. Plots indicate the copy numbers of the virus in each individual.

## Seasonal changes in environments, copepods, and viruses

We investigated the population dynamics of *P. newmani* and major viruses using zooplankton samples collected weekly over five months. Large environmental changes were observed during the sampling periods (Fig. 2a–e). Water temperature and salinity increased from March 18 (temperature: −0.3 °C; salinity: 32.3) to July 13 (temperature: 16.0 °C; salinity: 33.6) under the influence of the Soya Warm Current[28]. High nutrient concentrations (nitrate + nitrite and phosphate) were observed from March to early April, while low nutrient concentrations were observed after early April. After a nutrient decrease, chlorophyll fluorescence increased with increasing water temperature in April. The highest chlorophyll fluorescence was observed in late April, and sporadically high chlorophyll fluorescence values were observed until July.

*P. newmani*'s abundance presented clear seasonal changes (Fig. 2f) and demonstrated associations with the copy number and prevalence of the major viruses measured by RT-qPCR (Fig. 2g–j). Major viruses were only observed during *P. newmani*'s low-abundance period, and no major viruses were detected during *P. newmani*'s high abundance from March to early April. The major viruses in *P. newmani* exhibited two patterns of seasonal detection. PSNE-Pico1 and PSNE-Narna exhibited ≥ 90% prevalence in specific seasons. High copy numbers (average $3.8 \times 10^5$–$2.1 \times 10^7$ copies/ind., $n = 3$–10) and 100% prevalence were observed for PSNE-Pico1 after late June, which corresponded to a decrease in *P. newmani* abundance. High copy numbers and detection rates persisted until *P. newmani* seasonally disappeared in July. PSNE-Narna was detected in 90% of the *P. newmani* individuals on April 28 with an average of $3.5 \times 10^3$ copies/ind ($n = 10$). Contrastingly, the other two viruses PSNE-Pico2 and PSNE-Toga occasionally appeared in *P. newmani* throughout the study period and presented relatively low prevalence. PSNE-Pico2 was detected mainly from early April to early June and presented a prevalence of 10–50%. When a decline in copepod abundance was observed in early April, a maximum of $2.2 \times 10^8$

copies/ind was observed for PSNE-Pico2. PSNE-Toga was detected in late April and mid-June at a low prevalence (<15%), and up to $9.3 \times 10^4$ copies/ind were observed.

## Gene expression analysis

Gene expression analysis was performed to investigate the physiological changes in copepods with viral loads. A total of 763.5 M quality-filtered reads (15.4–38.4 M reads per sample) from 29 copepods (Supplementary Table 2) were assembled into 396,141 contigs and reduced to 241,006 contigs after clustering at 95% identity. Contigs derived from archaea, bacteria, fungi, and protozoa, including phytoplankton, were removed because they might be derived from parasitic, symbiotic and prey of copepods. The completeness analysis of the remaining 229,952 contigs demonstrated 88.2% complete, 7.1% fragmented, and 4.7% missing genes among the 1013 Arthropoda BUSCO.

We obtained an average overall alignment rate of 68.4% for each sample against the assembled contigs used for mapping. Based on the preliminary principal component analysis (PCA) results, two samples were excluded from further analysis as outliers (Supplementary Table 2). PCA of 27 samples demonstrated distinct gene expression patterns in copepods with PSNE-Pico1 viral loads (Fig. 3a). Although we observed seasonal changes in gene expression patterns in other samples, no clear gene expression patterns were observed in copepods with PSNE-Pico2, PSNE-Narna, and PSNE-Toga viral loads. Thus, we focused on PSNE-Pico1, and a total of 3816 upregulated and 1199 downregulated differentially expressed genes (DEGs) were detected in copepods with high PSNE Pico1 viral loads ($n = 3$) compared to the controls ($n = 4$) in June (Fig. 3b). Gene Set Enrichment Analysis (GSEA) of DEGs identified 54 enriched Kyoto Encyclopedia of Genes and Genomes (KEGG) pathways. KEGG pathways related to translation and degradation of protein, including ribosome (ko03010) and proteasome (ko03050), were prominent among the

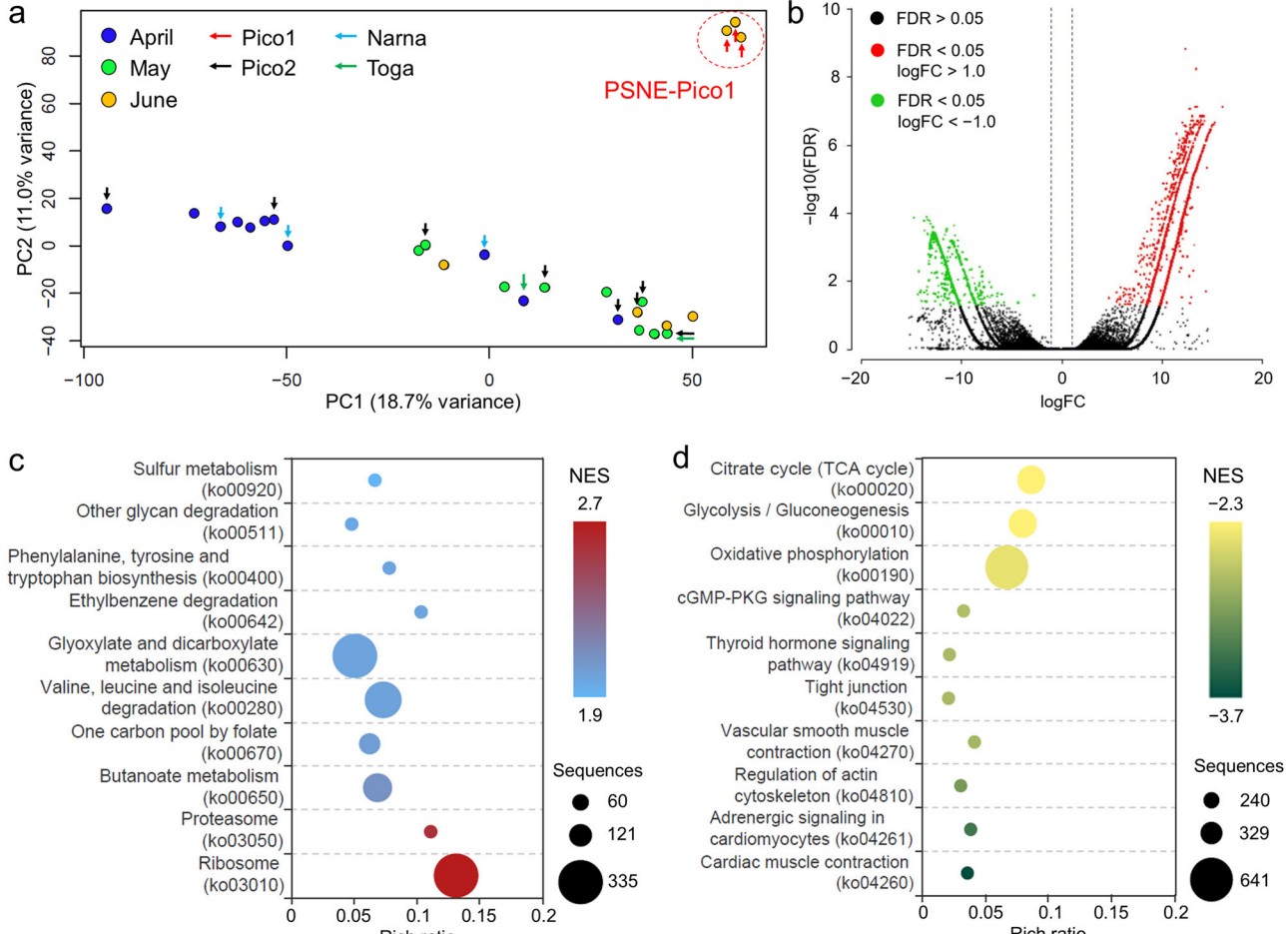

**Fig. 3 | Gene expression analysis of copepods. a** Principal component analysis (PCA) of the gene expression patterns. *Pseudocalanus newmani* individuals with high PSNE-Pico1 viral loads are surrounded by a red circle. **b** Volcano plot showing differentially expressed genes of *P. newmani* under high PSNE-Pico1 viral loads. Red and green colors indicate the upregulated and downregulated genes, respectively, under high PSNE-Pico1 viral loads. **c** Top ten over-represented enriched Kyoto Encyclopedia of Genes and Genomes (KEGG) pathways based on normalized enrichment score (NES) by Gene Set Enrichment Analysis (GSEA). **d** Top ten under-represented enriched KEGG pathways based on NES by GSEA.

over-represented pathways (Fig. 3c). Metabolism-related pathways were included in the top ten overrepresented KEGG pathways. Only the ribosome pathway (ko03010), which was over-represented in copepods with PSNE-Pico1, was selected as the Fisher's enriched pathway. In contrast, the abundant under-represented pathways were related to the circulatory system, including cardiac muscle contraction (ko04260), adrenergic signaling in cardiomyocytes (ko04261), and vascular smooth muscle contraction (ko04270), followed by pathways associated with cytoskeletal regulation of actin cytoskeleton (ko04810) and tight junctions (ko04530; Fig. 3d). Other under-represented pathways were associated with the endocrine system (thyroid hormone signaling pathway: ko04919), signal transduction (cGMP-PKG signaling pathway: ko04022), and metabolism (oxidative phosphorylation, ko00190; glycolysis/gluconeogenesis, ko00010; citrate cycle, ko00020). These KEGG results were consistent with the enriched Gene Ontology (GO) terms (Supplementary Table 3), with the top ten GO terms mostly assigned to the biological process category for upregulated genes and the cellular component category for downregulated genes.

**Transmission electron microscopy analysis**
Using transmission electron microscopy (TEM), we focused on copepods during the high viral load period of PSNE-Pico1 to observe viral particles based on the results of phylogenetic, population dynamics, and gene expression analyses. We observed the aggregation of virus-like particles (VLPs) in the gut contents (Fig. 4a). Although the morphology of the gut contents was often damaged, VLPs were observed in the phytoplankton-like

particles. We also observed the possible presence of VLPs around the microvilli of copepod intestines (Fig. 4b). Additionally, we observed VLPs aggregating inside the copepod intestines (Fig. 4c). The diameter of the viral particles was $31.7 \pm 2.3$ nm (mean $\pm$ SD, $n = 8$) in gut contents and $30.7 \pm 1.8$ nm ($n = 8$) in copepod intestines.

**Genetic population structures**
Mitochondrial cytochrome c oxidase subunit I (mtCOI) sequences of *P. newmani* were analyzed to clarify whether viral detection was associated with seasonal changes in the genetic population structure of copepods. In total, 65 mtCOI haplotypes (619 bp) were detected in 143 *P. newmani* individuals throughout the sampling period ($n = 18$–48 per month). No clear haplotype and genetic diversity changes were observed throughout the sampling periods (haplotype diversity (Hd): 0.90–0.94; nucleotide diversity ($\pi$): 0.0099–0.123; Supplementary Table 4). The major *P. newmani* haplotypes were seasonally shared among *P. newmani* during different months (Fig. 5a). Additionally, no genetic structural changes based on mtCOI were indicated by pairwise $\Phi_{ST}$ distances, and significant differences were not observed between sampling months (Supplementary Table 5).

**Eukaryotic parasites, predators, and prey of copepods**
Predators and other eukaryotic parasites in addition to viruses have been investigated as possible factors associated with copepod population dynamics. Regarding the seasonal changes of *P. newmani*'s planktonic predators, we focused on medusa of Hydrozoa because the abundance of

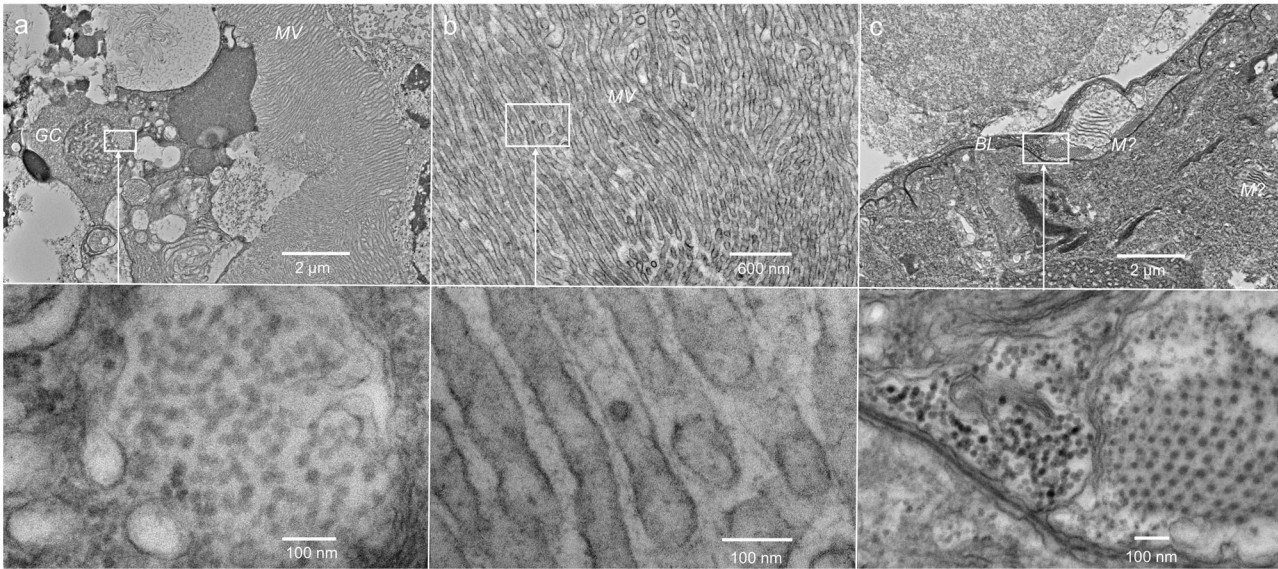

**Fig. 4 | Transmission electron microscopy (TEM) images of virus-like particles (VLPs) in *Pseudocalanus newmani*. a** VLP aggregation in the gut contents. **b** VLPs around the microvilli. **c** VLP aggregation in the copepod intestine. Basal lamina (*BL*), gut content (*GC*), microvilli (*ML*), mitochondria (*M*).

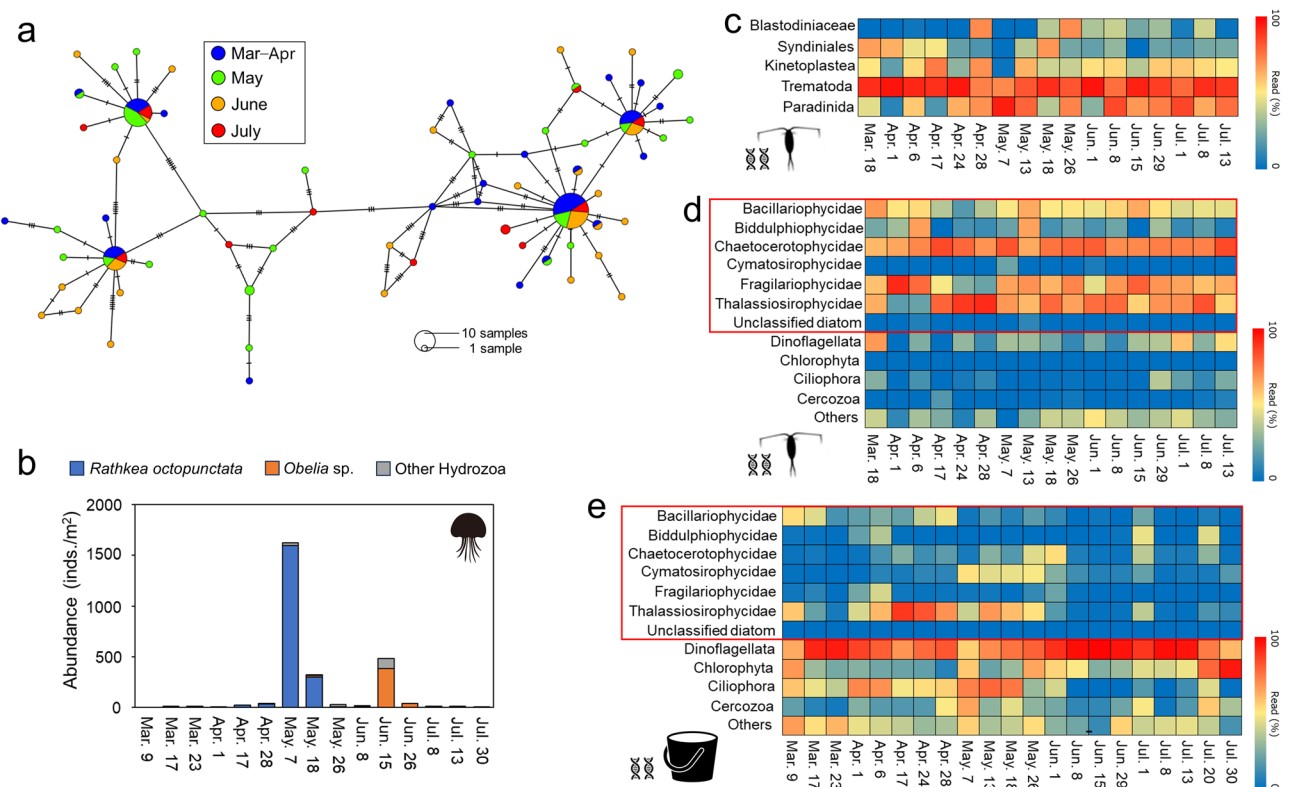

**Fig. 5 | Genetic population, predator, eukaryotic parasite, and prey of copepods. a** Minimum spanning networks for the haplotypes in each month using mitochondrial cytochrome c oxidase subunit I (mtCOI) sequences. The balloon sizes indicate the number of individuals in each haplotype, and the lines between haplotypes represent the sequence differences in mtCOI. **b** Abundance of major

other possible predators, including chaetognaths, ctenophores, siphonophores, scyphozoans, and fish larvae, was low in our samples collected using small plankton nets. *Rathkea octopunctata* and *Obelia* sp. are two abundant hydrozoan medusae species. The highest *R. octopunctata* abundance (1599.3 individuals/m³) was observed in early May (Fig. 5b), while the highest *Obelia* sp. abundance (380.3 individuals/m³) was observed in mid-June.

Hydrozoa species as predators of copepods. Results of 18S metabarcoding for (**c**) eukaryotic parasites in *Pseudocalanus newmani*, (**d**) protist communities obtained from *P. newmani*, and (**e**) protist communities in the ambient waters. Heatmaps indicate average proportions of sequence reads (log-transformed) on each sampling day. Red lines indicate the detailed taxonomy of the diatom for (**d**) and (**e**).

A total of 2010–22,011 (average 5527, $n = 21$) reads in the environmental plankton communities and 1542–54,283 (average 14,252, $n = 67$) reads in the copepods were obtained after quality filtering using 18S metabarcoding, and they were classified into 64 parasitic and 129 non-parasitic operational taxonomic units (OTUs). The sequence proportions of parasite OTUs were low in the environmental samples (0–2.0%; average

**Table 1 | Best model of variables explaining copepod abundance**

| Variable | Estimate | Z value | *P* |
|---|---|---|---|
| Intercept | 7.67 | 16.13 | < 0.001 |
| Water temperature | −0.13 | −3.29 | < 0.005 |
| Virus prevalence | −1.25 | −2.94 | < 0.005 |
| Diatom proportion | −4.86 | −4.55 | < 0.001 |

The variables were selected using a stepwise method in a generalized linear model analysis based on a pseudo-$R^2$ value of 0.77 and Akaike information criterion of 209.16.

0.3%) and medium-high in the copepod samples (5.7–99.4%; average 65.8%). Eukaryotic parasitic taxa in *P. newmani* exhibited no clear seasonal patterns, while Trematoda demonstrated high proportions (25.2–82.1%; average 63.5%) throughout the sampling period (Fig. 5c). Relatively high Syndiniales proportions were observed from March to mid-April, and the proportion of Paradinida increased to 74.7% from April 24 to mid-July. Other major parasitic eukaryotic organisms included Kinetoplastea and Blastodiniaceae.

Non-parasitic OTUs represented the possible prey of copepods, and high diatom read proportions (71.0–99.4%; average 91.2%) were detected in the copepod samples (Fig. 5d; Supplementary Fig. 2). Major diatom OTUs exhibited a high proportion of the diatom family Fragiariaceae until early April. High Thalassiosirfaceae and Chaerocerotaceae OTU proportions were observed between mid-April and mid-July. No clear changes in diatom prey were observed between June 15 and 29, when the PSNE-Pico1 loads dramatically changed. Additionally, the environmental samples showed relatively high diatom proportions (0.5–62.0%; average 17.5%), although other phytoplankton, including dinoflagellates, were dominant (12.3–95.8%; average 50.6%) through the sampling period (Fig. 5e; Supplementary Fig. 2). Seasonal changes in diatom proportions were observed between the environmental samples and *P. newmani*, including clear peaks of high Thalassiosiraceae proportions (up to 57.9%) in late April.

### Variables explaining the population dynamics of copepods

To evaluate the factors influencing copepod abundance, a generalized linear model (GLM) analysis was performed. The best-selected model included average water temperature, viral prevalence, and diatom proportions in ambient water (Table 1). All three variables were negatively correlated with the changes in copepod abundance ($P < 0.005$, $n = 17$), indicating that the virus could improve the dynamics of copepod abundance, in addition to water temperature and food quality. The best model did not include other factors such as salinity, chlorophyll fluorescence, nutrients, eukaryotic parasites (Syndiniales), and predators (Hydrozoa).

### Discussion

The Earth's environment and ecosystems have been rapidly changing under climate change, and increasing temperatures may lead to a high risk of disease by pathogens[29]. Zooplankton and viruses are key players in marine ecosystems; however, their interactions have been poorly investigated. Although previous studies on viruses associated with zooplankton focused on detecting viruses[15,17,19,30], to the best of our knowledge, the present study is the first to compare the detailed population dynamics between copepods and viruses using samples collected weekly. Integrated approaches involving morphology and multiple molecular methods detected four novel RNA viruses in the copepod *P. newmani* with different seasonal patterns, which were possibly associated with transmission patterns. As hypothesized, the impacts of these viruses on copepod abundance and physiology were demonstrated, suggesting the importance of zooplankton viruses for further understanding of marine ecosystems.

Among the four major viruses, we observed two detection patterns, although seasonal changes in copy number and prevalence varied for each virus. One pattern included PSNE-Pico1 and PSNE-Narna, which demonstrated high prevalence (≥90%) at a specific period. Phylogenetic

analyses indicated that these viruses were related to taxonomic groups of phytoplankton (*Marnaviridae*)[23] and fungal viruses (*Narnaviridae*)[31]; however, viruses related to these families were recently detected in the transcriptome data of plants and invertebrates (Fig. 1). PSNE-Pico1 was phylogenetically related to the diatom virus *Bacillarnavirus*, agreeing with the detection of diatoms as the major prey of *P. newmani* via 18S meta-barcoding. Additionally, we detected PSNE-Pico1 and PSNE-Narna in the copepod *Acartia* spp. on the same sampling day (Supplementary Fig. 3), and PSNE-Pico1 was not detected from filtered seawater in March but from that in July (Supplementary Fig. 4). These results indicate the prey of *P. newmani* is a source of some viruses. Although *P. newmani* after incubation in filtered seawater was used to remove gut contents before obtaining viral sequences using transcriptome data in this study, phytoplankton viruses vectored by copepods tend to be retained in guts beyond a regular gut clearance time of copepods[10,32]. Similar sizes of VLPs were observed between the gut contents and copepods; however, it is not possible to conclude if these VLPs are same viruses only based on the TEM analysis. Therefore, horizontal transmission and propagation of viruses in copepods should be carefully discussed. Another group included PSNE-Pico2 and PSNE-Toga, and these viruses mainly demonstrated low prevalence (≤50%) at longer periods from April to June. PSNE-Pico2 formed a new genetic lineage with other invertebrate viruses, indicating that these viruses are important when considering the evolution of RNA viruses. Neither virus was detected among *Acartia* spp. and filtered seawaters (Supplementary Fig. 3). Thus, they may be vertically transmitted copepod viruses. In the present study, we observed other possible DNA and RNA viruses associated with *P. newmani* from the transcriptome data; however, they were mostly rare or undetectable via qPCR. Additionally, VLPs in phytoplankton-like gut contents were detected from the nucleus, which is a characteristic of the ssDNA virus of marine diatom, raising a possibility of a different virus from *Marnaviridae* virus, which replicates in the cytoplasm[23,33]. Although the number of major viruses detected in this study was within the range of other invertebrate viruses (1–6 high-frequency viruses)[24], further investigations may lead to the detection of additional DNA and RNA viruses associated with *P. newmani*.

All major viruses were detected only during the low-abundance period of copepods with different seasonal peaks. Particularly, after a decline in copepod abundance, the highest PSNE-Pico2 viral loads were observed in April. Additionally, the 100% PSNE-Pico1 prevalence persisted until *P. newmani*'s seasonal disappearance in July. The copy numbers of the viruses were almost equivalent to or sometimes exceeded those of the ssDNA viruses (>$10^5$ copies per individual) reported in another coastal copepod, *Labidocera aestiva*[15]. Although no studies have investigated the seasonality of copepods and associated viruses in marine ecosystems, viruses have been suggested as the controlling factors for Arthropoda population dynamics in freshwater ecosystems[21]. In addition to water temperature and food quality (diatom proportions), we demonstrated that viruses were among the factors controlling *P. newmani*'s population dynamics. Viruses might have been activated with an increase in water temperature and productivity because large copepod abundances existed during low-temperature periods. In the coastal marine ecosystems around Japan, the occurrence of copepod carcasses increases with water temperature[34]. Despite the ecological importance of the association between viruses and copepod population dynamics, limited data are available on this association. Since we demonstrated that viral prevalence can increase the accuracy of predicting the population dynamics of major zooplankton species, further sampling efforts of zooplankton viruses are urgently required to understand the changes in marine ecosystems under the global warming scenario.

In addition to viruses, other factors associated with copepod abundance were considered in this study. For example, diatom blooms are thought to deleteriously affect *P. newmani* reproduction[35,36]. These diatoms include *Chaetoceros* and *Thalassiosira* species, which were mainly dominant during spring and detected as the major foods of *P. newmani* throughout the sampling period in this study. Food quality is an important factor in *P. newmanii*'s population dynamics because the diatom proportion was selected as the best model explaining copepod abundance. Our 18S

metabarcoding analysis revealed high eukaryotic parasite diversity. Some parasites, including members of Syndiniales, are lethal in some cases[37]. However, their proportions were low in the parasitic reads, and their seasonality related to copepod abundance was unclear. Given that the interactions between planktonic copepods and eukaryotic parasites, including the major taxa of Trematoda, have not been fully characterized, future studies that include eukaryotic, prokaryotic, and viral parasites would be of interest for better understanding copepod ecology. Carnivorous zooplankton taxa include *R. octopunctata* and *Obelia* sp., which feed on copepodites and the nauplii of copepods[38,39]. Occasional peaks of these hydrozoan predator abundances after the spring bloom may impact on copepod abundance. Finally, genetic populations did not clearly change throughout the sampling periods, as supported by a more sensitive genome-wide method in the same study area[40], and *P. newmani* migrations were not a significant factor in the population dynamics in our study area.

In addition to the possible influence of viruses on copepod abundance, viruses were associated with clear physiological changes in copepods, especially during the high prevalence of PSEN-Pico1. PSEN-Pico1 exhibited high copy numbers and prevalence until the copepods seasonally disappeared in the study area. Substantially upregulated ribosome-associated genes may suggest viral replication and protein synthesis in copepods, whereas protein degradation by viral infections has been suggested in copepod hosts by gene expression analyses, which were performed after removing possible contamination by protistan data, including phytoplankton. The same patterns of changes in KEGG pathways related to ribosome and proteosome pathways have been reported in *Medusavirus*-infected amebas[41]. Additionally, downregulated genes associated with the cytoskeleton, muscle, and circulatory systems are suggested to be the cause of reduced copepod movement activities. This reduced activity may lead to higher predation pressure, and interactions between copepods and viral infections were inferred from the preliminary analysis of copepods[16]. Additionally, we observed upregulated functions associated with the immune system. For example, butanoate metabolism plays a role in immunity against antigens in the intestine[42]. There are persistent propagative/non-propagative plant viruses in insect vectors in terrestrial ecosystems, and they may also be pathogenic and cause transcriptomic responses of insect vectors, including immune responses in invertebrates[13]. Our results demonstrated that the presence of viruses ingested together with phytoplankton caused large physiological changes in *P. newmani*, although influential viruses and their locations, propagations, and transmissions within copepods should be carefully evaluated. There are also possible indirect effects of phytoplankton viruses by degrading copepod food sources because viral infections release nutrients through cell lysis, decreasing carbon transfer efficiency from lower to higher trophic levels[2]. In addition to understanding the physiological changes induced by other major viruses, the physiological changes suggested by transcriptome analysis should be validated in incubation experiments or natural environments.

This study provides a first step toward understanding the ecological roles of planktonic copepod-associated viruses, and our results raise various questions about the functions of zooplankton viruses in marine ecosystems. First, the possible impact of viruses on population dynamics and physiological changes in copepods was indicated; however, direct evidence of copepod death associated with viral infections was not obtained. To clarify viral propagations, incubation experiments should be conducted to monitor viral loads using copepods after digesting gut contents completely. Considering that copepod carcasses alter biogeochemical cycles in oceans[9], viral loads should be compared between live and dead copepods to investigate whether viral infection leads to the natural death of copepods. Second, the detailed transmission patterns of viruses in copepods must be clarified because this study showed that some viruses are ingested by copepods with their food sources. We focused only on adult female copepods in this study; however, viruses should be investigated in adult males, other developmental stages of copepods (including eggs), environmental water, and phytoplankton. Additionally, other approaches, such as fluorescence in situ

hybridization, strand-specific transcriptome, and microRNA analyses, can be useful for studying the detection of a specific virus, virus localization and replication and host responses to pathogens. Third, the visualization of copepod behavior may reveal alterations under viral infection that influence copepod movements. Although the possible importance of zooplankton viruses has been recognized[20], our results provide strong evidence for the importance of investigating the interactions between copepods and viruses. Given that copepods and viruses are ubiquitous and abundant in oceans, further studies covering various temporal and spatial areas are expected to fill the gap in our understanding of marine ecosystems.

## Methods

### Sampling

Zooplankton samples were collected almost weekly from March 18 to July 20, 2020, at the Okhotsk Tower, located 1 km off the coast of Mombetsu, northeastern Hokkaido, Japan (Supplementary Fig. 1; 44° 20.2' N, 143° 22.9' E). No *P. newmani* were observed after July 20. Using a NORPAC net with a 335-μm mesh size, all samplings were conducted during the daytime. A vertical tow was performed from the bottom (approximately 10 m in depth) to the surface. The samples were immediately preserved in RNAlater and kept at −20 °C until genetic analyses. Additionally, the samples for morphological classification were collected using the same method and preserved in a 5% formalin solution. Vertical profiles of water temperature, salinity, and chlorophyll fluorescence were measured daily during the sampling period using a RINKO-Profiler ASTD102 (JFE Advantech Co., Ltd.). The nitrate + nitrite and phosphate nutrients were measured by collecting surface water using a bucket on the same day as the zooplankton sampling.

### Virus detection from copepods

Additional samples using the plankton net were collected on March 18 and June 30 to create a virus catalog. Adult female *P. newmani* were identified morphologically from bulk zooplankton samples under a stereomicroscope and preserved in RNAlater after approximately 6 h incubation at environmental water temperature in filtered sea waters using a 10-μm mesh size net to depurate the copepod guts. A total of 100 *P. newmani* individuals were used for RNA extraction using a Direct-Zol RNA MiniPrep Kit (Zymo Research). RNA quality was assessed using the 4200 TapeStation (Agilent Technologies), and RNA concentration was measured using a Qubit 3.0 Fluorometer (Life Technologies). After rRNA removal, a sequence library was prepared using the MGIEasy rRNA Depletion Kit, MGIEasy RNA Directional Library Prep Set, MGIEasy Circularization Kit, and High-throughput Sequencing Kit (MGI Tech Co., Ltd.). Sequence data of 2 × 150 bp paired-end were obtained using DNBSEQ-G400 (MGI Tech Co., Ltd.). Raw sequence data were quality-filtered using Trimmomatic[43] with default settings, and the rRNA sequences were removed using SortMeRNA[44]. Quality-filtered short reads were assembled into contigs at a minimum length of 300 bp using Trinity[45]. Possible viral contigs were identified based on the results of DIAMOND BLASTX[46] by comparing with the GenBank nonredundant (nr) protein database (downloaded in September 2022), with an e-value cut-off of $1 \times 10^{-5}$. The contigs with BLAST hits to RdRp (RNA viruses) were selected. Quality-filtered reads were mapped back to the contigs using Bowtie2[47]. Herein, we selected the major contigs of possible viruses with > 1000 bp and 100 mapped reads, as confirmed by preliminary qPCR analysis. We used NCBI ORFfinder and a conserved domain search to predict ORF and gene positions for the major viruses. The phylogenies of the major viruses were further analyzed, and the amino acid sequences of the RdRp gene were aligned using MUSCLE[48]. Ambiguous positions were excluded using trimAl version 1.4[49] with the gappyout option, and the best substitution model for the amino acid sequences was selected based on the AIC in ModelTest-NG[50]. Phylogenetic analysis was performed using 100 bootstrap replicates of RAxML-NG[51]. Notably, two major contigs were identified with the highest BLASTX hits to the replication-associated proteins of the circular ssDNA viruses and another major contig of the Tombus-like ssRNA virus. These viruses were

not detected in individual copepods in our preliminary qPCR analyses and were excluded from the major viruses.

## Population dynamics of copepods and viruses

The abundance of adult female *P. newmani* in the RNAlater samples was determined to investigate changes in *P. newmani* abundance during the sampling period. All adult female individuals were picked up and identified morphologically under a stereomicroscope from each zooplankton sample, and ten adult females were selected in the case of ≥ 10 *P. newmani* individuals. DNA (total 30 μL) and RNA (total 20 μL) were extracted from each individual using AllPrep DNA/RNA Micro Kit (Qiagen). Genomic DNA was used to investigate the genetic population structures and gut contents of *P. newmani*, as described below. In contrast, RNA was used to calculate the viral loads or analyze the gene expression of *P. newmani*.

For seasonal changes of viruses, total RNA (2.0 μL) after removing genomic DNA using DNase I (Invitrogen) was reverse-transcribed into first-strand cDNA (a total of 20 μL) using SuperScript IV Reverse Transcriptase (Invitrogen). Primers and probes were constructed for each major virus (Supplementary Table 6), and the copy numbers of the viruses in each *P. newmani* individual were measured using RT-qPCR. RT-qPCR was performed in 20 μL reaction mixtures containing 4.0 μL of distilled water, 10 μL of Probe qPCR Mix (TaKaRa), 1.6 μL of each primer (5 μM), 0.8 μL of probe (5 μM), and 2.0 μL of template cDNA using LightCycler 480 II (Roche). The templates used were duplicates of cDNA and triplicates of $10$–$10^8$ copies μL$^{-1}$ of standards. PCR cycling was conducted for each virus as follows: 2 min denaturation at 95 °C, followed by 45 cycles of 5 s denaturation at 95 °C, and 30 s annealing and extension at 60 °C. Seasonal changes in viral loads and the prevalence of each virus were compared with the environmental conditions and *P. newmani* abundance. Additionally, we also carried out RT-qPCR for each virus using copepod *Acartia* spp. to evaluate whether the detected viruses were unique to *P. newmani*.

## Physiological changes in P. newmani

We selected 29 adult *P. newmani* females from the population dynamics samples for transcriptome analysis to evaluate the physiological changes in *P. newmani* based on the detection of major viruses (Supplementary Table 2). After checking the RNA quantity and quality as described above, total RNA was purified using the NEBNext Poly(A) mRNA Magnetic Isolation Module (New England Biolabs), and the sequencing libraries were constructed using the TaKaRa SMART-Seq Stranded Kit (TaKaRa). We obtained 2 × 150 bp paired-end sequences by sequencing on the Illumina HiSeq platform, and bioinformatics analyses were performed using the OmicsBox (https://www.biobam.com/omicsbox/). The raw sequence data were quality-filtered using Trimmomatic with the following options: LEADING:20, TRAILING:20, AVGQUAL:25, and MINLEN:50. We performed SortMeRNA and Trinity analyses, and the assembled contigs (minimum 300 bp) were clustered at a 95% similarity threshold using CD-HIT-EST v.4.8.1[52] to avoid high multiple alignment rates in the completeness assessment and mapping rates. Contigs were taxonomically classified using Kraken 2.1.2[53], and those identified as archaea, bacteria, fungi, or protozoa were removed. The completeness of the de novo assembly was assessed using BUSCO v.4.05[54] with the default settings using the Arthropoda database. Contigs were annotated based on the results of the DIAMOND BLASTX search against the GenBank nr protein database, with an e-value cut-off of $1 \times 10^{-5}$. GO terms were retrieved for contigs based on BLAST hits using Blast2GO[55]. Additionally, functional annotation was performed using EggNOG-Mapper 2.1.0 with EggNOG 5.0.2[56], merged with the Blast2GO results.

All clean reads were mapped to the contigs using Bowtie2 v.2.3.4.3[47], and the expression level was quantified for each contig using RSEM v.1.3.1[57]. PCA was performed to evaluate gene expression patterns based on $\log_{10}$-transformed counts per million (CMP) reads using R. Contigs with <1 CPM at any samples were excluded for PCA. DEGs between control and high viral loads were identified using edgeR v.3.28.0[58] with a |log(fold-change)| ≥ 1 and false discovery rate (FDR) ≤ 0.05. Enrichment analysis was conducted for the DEGs using Fisher's exact test (FDR ≤ 0.05) and GSEA[59] to detect over- or under-represented GO terms and KEGG pathways[60].

## TEM analysis

Additional *P. newmani* specimens were collected on July 2, 2022, to confirm the presence of viruses in the copepods. We used the same sampling method for analyzing population dynamics, and RT-qPCR demonstrated 100% PSNE-Pico1 prevalence in *P. newmani* on the sampling day. In addition to sampling for genetic analysis, live specimens of adult *P. newmani* females were immediately sorted out from bulk zooplankton, fixed in 2% glutaraldehyde and 2% paraformaldehyde in 0.1 M cacodylate buffer, and preserved at 4 °C. TEM analysis was performed at the Hanaichi Ultrastructure Research Institute (Aichi, Japan). The pre-fixed *P. newmani* samples were rinsed in 0.1 M cacodylate buffer overnight at 4 °C and post-fixed in 2% osmium tetroxide for 2 h at 4 °C. The samples were subsequently dehydrated in 30%, 50%, 70%, 90%, and 100% ethanol and embedded in epoxy resin. Ultrathin (80–90 nm) sections were obtained using an ultra-microtome, stained with 2% uranyl acetate for 15 min, and stained with lead solution for 2 min. The stained sections were examined by TEM using a HITACHI H-7600 microscope operated at 100 kV. TEM images were analyzed to measure the diameters of the VLPs using ImageJ (https://imagej.net/software/imagej/).

## Genetic population structures of P. newmani

The MtCOI sequences were obtained using DNA from each individual to investigate the seasonal changes in genetic population structure. We used primer pairs LCO1490_Pseudocalanus and HCO2198_Pseudocalanus (Supplementary Table 6)[40]. PCR amplification was performed in 15 μL reaction mixtures containing 3.5 μL of distilled water, 1.5 μL of each primer (5 μM), 7.5 μL of KOD One PCR Master Mix (TOYOBO), and 1.0 μL of template DNA. PCR cycling was conducted as follows: 1 min denaturation at 94 °C, followed by 35 cycles of 10 s denaturation at 98 °C, 5 s annealing at 45 °C, 5 s extension at 72 °C, and a final extension at 68 °C for 1 min. DNA amplification was confirmed via electrophoresis on a 2% agarose gel. All PCR products were purified using ExoSap-IT (Applied Biosystems) and prepared for sequencing using a BigDye Terminator v3.1 Cycle Sequencing Kit (Applied Biosystems). Sanger sequencing was performed using a 3130 DNA Sequencer (Applied Biosystems). All obtained sequences were analyzed, and the haplotype numbers, Hd, and π were calculated for each month using DnaSP 6.12.03[61]. Pairwise $\Phi_{ST}$ distances were calculated between months with 1,000 permutations using ARLEQUIN 3.5.2.2[62], and significant differences were assessed after Bonferroni correction. Minimum spanning networks for haplotypes were obtained using PopART 1.7[63].

## Planktonic copepod predators

Specimens preserved in 5% buffered formalin were classified as the lowest-ranking taxon under a stereomicroscope. The abundance (individuals/m$^3$) of each organism was counted, and seasonal changes in the major copepod predators were investigated in samples collected using the plankton net.

## Eukaryotic parasites and prey sources of P. newmani

For the prey and eukaryotic parasites of *P. newmani*, 18S metabarcoding was performed on ambient water and copepods. Seawater samples were collected weekly from the sea surface using a bucket. Seawater (750-1982 mL) was filtered on 0.22-μm Sterivex filters. Genomic DNA from the filtered samples was extracted using 5% Chelex buffer[64]. The 18S V7-V9 region (approximately 500 bp) was amplified using KOD Plus version 2 (Toyobo) and the universal primer pair 18S-V7F and 18S-V9R with adaptor sequences (Supplementary Table 6)[65]. Each PCR sample was prepared in 25 μL reaction mixture containing 12.0 μL of distilled water, 2.5 μL of 10×buffer, 2.5 μL of dNTPs (2 mM), 1.5 μL of MgSO$_4$ (25 mM), 2.5 μL of each primer (10 μM), 0.5 μL of KOD Plus polymerase, and 1.0 μL of template DNA. PCR cycling included initial denaturation at 94 °C for 3 min, followed by 30 cycles of 15 s denaturation at 94 °C, 30 s annealing at 56 °C, and 40 s extension at 68 °C. Additional adaptor and index sequences to

discriminate the samples were attached during the second round of PCR. The first PCR products were purified using AMPure XP (Beckman Coulter) and used for the second PCR. The second PCR was carried out in the same way as the first PCR with 12 PCR cycles.

The genomic DNA of four *P. newmani* individuals, which were used for population genetic analysis, was selected on each sampling day for 18S metabarcoding analysis of copepods. The two-step PCRs used for seawater samples did not work for all the copepod samples. Thus, we used 18S-V7F and 18S-V9R primers without adaptor sequences in the first PCR for the copepod samples, as performed in other metabarcoding diet analysis[66]. In the first PCR of copepods, peptide nucleic acid (PNA) was used to block the amplification of the host sequences (Supplementary Table 6). The first PCR was performed in 15 μL reaction mixture containing 5.5 μL of distilled water, 1.5 μL of 10×buffer, 1.5 μL of dNTPs (2 mM), 0.9 μL of MgSO$_4$ (25 mM), 0.9 μL of each primer (5 μM), 1.5 μL of PNA (10 μM), 0.3 μL of KOD Plus polymerase, and 2.0 μL of template DNA. The first PCR was conducted as follows: 2 min denaturation at 94 °C, followed by 30 cycles of 10 s denaturation at 98 °C, 30 s PNA annealing at 66 °C, 30 s primer annealing at 56 °C, 1 min extension at 68 °C, and a final extension at 68 °C for 7 min. The adaptor and index sequences were attached during the second and third PCRs. The second and third PCRs without PNA were conducted in the same way as the first PCR, and PCR cycles were set at 8 cycles with annealing temperature at 50 °C for second PCR and 59 °C for the third PCR. The PCR products were purified and diluted three-fold using AMPure XP after each PCR step.

All PCR product libraries were sequenced using the MiSeq Reagent Kit v3 (600 cycles) on an Illumina MiSeq, and 2 × 300 bp paired-end sequence reads were obtained. Raw sequence data were initially quality-filtered using Trimmomatic. Sequence reads of ≥ 250 bp remained after removing the adaptor sequences. We cut sequence bases with quality scores ≤ 20 at the 5′ and 3′ ends, and reads with average quality scores ≥ 30 for every 30 bp were used for further bioinformatics analysis in mothur v.1.47.0[67]. Paired-end sequences were merged, and only reads with both primer sequence regions were retained. After removing the primer sites, we retained only reads with no ambiguous bases (Ns) and < 7 homopolymers. Quality-filtered reads were aligned using the reference-aligned sequences in SILVA v132[68]. We retained only reads with ≥ 400 bp length, which were properly aligned to the 18S V7-V9 regions. (start: 75%; end: 25%). All reads were pre-clustered based on 4 bp differences at the most, and the possible chimera sequences detected using UCHIME[69] and singletons were removed. Taxonomic classification of reads was performed by a naïve Bayesian classifier[70] using the PR$^2$ database version 4.14.0[71]. In this study, we mainly focused on eukaryotic parasites and protist copepod prey, and reads classified as non-Eukaryota and Opistholonta were removed, except for Trematoda and Choanoflagellida. The remaining reads were clustered into OTUs using a 97% similarity threshold. We only retained OTUs found for at least two samples and with ≥ 1% sequence reads in any samples. Taxonomic information was obtained using an NCBI BLAST search against the nucleotide database. We separately analyzed copepod parasites and prey based on the results of the taxonomic classification.

## GLM analysis
The variables influencing copepod abundance were evaluated using GLM analysis in R. The initial parameters of the variables included water temperature, salinity, chlorophyll fluorescence, nutrients, viruses, eukaryotic parasites, predators, and diatoms. For environmental parameters, phosphate in the surface layer was used as a nutrient value, in addition to the average water temperature, salinity, and chlorophyll fluorescence. The virus parameter was determined based on the proportion of individuals in which any virus was detected. The parasite and predator values used were the Synidiniales proportions in parasitic taxa determined using 18S metabarcoding and Hydrozoa abundance determined using morphological analysis, respectively. The diatom parameter represents the proportion of diatom reads in the 18S metabarcoding of ambient water. Based on the results of the dispersion tests, a negative binomial distribution with the log

link function was used for the GLM analysis. The best model was selected based on the AIC values obtained through backward selection of effective variables. Furthermore, the goodness of fit for the best model was estimated based on McFadden's pseudo R$^2$.

## Statistics and reproducibility
Statistical analyses of DE and enrichment analysis for the gene expression analysis were conducted using the standard workflow in the OmicsBox. The GLM analysis was performed in R. Details of the sample size, numbers of replicates, and statistical analyses for each experiment are listed in the respective parts of the Results and Methods sections.

## Reporting summary
Further information on research design is available in the Nature Portfolio Reporting Summary linked to this article.

## Data availability
Data of the main results, including source data for graphs, are available at Figshare (https://doi.org/10.6084/m9.figshare.c.7389289). The mtCOI sequences used are available in GenBank (accession numbers: LC792395–LC792537). All raw sequence data obtained using high-throughput sequencing are available from the NCBI/EBI/DDBJ Sequence Read Archive (BioProject accession number PRJDB17408). The sequences of assembled contigs for major viruses are available in GenBank (accession numbers: LC810156–LC810159).

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

## Acknowledgements
The authors thank Hitomi Yoshida for providing assistance with sampling and Mieko Noda for providing assistance in the laboratory. This work was supported by the Japan Society for the Promotion of Science (grant number 20H03057) and the Environment Research and Technology Development Fund (JPMEERF20224R03) of the Environmental Restoration and Conservation Agency provided by the Ministry of the Environment of Japan.

## Author contributions
J.H. designed the study and performed data analyses. S.K. conducted field sampling. H.K. provided the environmental data. S.N. provided advice on the molecular analysis and obtained metabarcoding data for ambient waters. J.H. wrote the manuscript. All authors revised the manuscript and approved its final submission.

## Competing interests
The authors declare no competing interests.
