## [Transparent Peer Review file · Communications Biology]

Ecological interactions between marine RNA viruses and planktonic copepods

Corresponding Author: Dr Junya Hirai

Version 0:

Reviewer comments:

Reviewer #1

(Remarks to the Author)

This paper focuses on the interaction between zooplankton and viruses, investigating the ecological role of viruses in the zooplankton copepod *Pseudocalanus newmani*. It analyzes transcriptome data from *P. newmani* and discovers four novel RNA viruses, and further mentions that some of these viruses regulate the population dynamics of *P. newmani*. This is the first study to detail the impact of viruses on marine zooplankton, providing an important perspective on understanding marine ecosystems. However, there are some extremely important issues in this paper that need to be revised.

The authors' discussion on the possibility of horizontal transmission of the viruses detected in this study (PSNE-Pico1 and PSNE-Narna) is not scientific and needs more thorough verification. In their discussion, they mention that viral infection of *P. newmani* has occurred because the virus was detected when *P. newmani* was cultured in filtered seawater (L243-245), but it is unclear how it was cultured. Demonstrating whether the gut contents were completely digested during that incubation period, or showing that an increase in viral genome occurred even after the gut contents were gone, would indicate that virus infection of *P. newmani* had occurred, but such data were not provided. With these insufficient evidences, I believe that PSNE-Pico1, which was particularly numerous, only detected viruses infecting the feeding diatoms, making horizontal transmission doubtful.

Virus-like particles have also been observed in *P. newmani* tissue microvilli in the gut tract (Fig. 4b). It is extremely important to verify whether these particles are viruses. It is necessary to measure and verify the size of the VLPs in multiple images to objectively determine if the size of these particles is roughly the same as the virus-like particles found in the cells of the feeds, which are also present in the gut tract. Furthermore, it is not possible to determine from this image whether VLPs are really present in the microvilli. In this study, TEM observations of 80-90 nm thick sections were observed, and when a typical diatom virus is around 30 nm and the virus particles are on the surface of the microvilli, if they are observed in a single section, then VLPs within the VLPs could appear to be present within the microvilli. To observe such a small VLP by TEM and confirm its presence within the microvilli, it would need to be observed three-dimensionally, e.g. using tomography. As mentioned above, it may be difficult to consider the results shown alone from the perspective of horizontal transmission of *P. newmani* by a virus that infects the feed. To show that the virus is infecting *P. newmani*, it is important to confirm viral amplification in the cells and show data on physiological increases from the initial viral load. The authors mention that there are no data directly showing these (L314-415), so discussions based on horizontal transmission should be avoided as much as possible. On the other hand, it is possible, for example, that the feeding plankton may have indirectly affected *P. newmani* physiology by decreasing its feeding quality as a result of the viral infection. In addition to horizontal transmission, these multifaceted possibilities should be equally mentioned.

The authors state that VLPs of gut tract contents are observed in the nuclei of phytoplankton cell-like particles (Fig. 4a, L167-168); as this sample is a *P. newmani* sample from a time when PSNE-Pico1 was detected in large numbers in M&M, the authors suggest that the VLPs in the gut contents are PSNE-Pico1 particles. On the other hand, diatom RNA viruses multiply in the cytoplasm and not in the nucleus, so if this VLP is in the nucleus, it is a different virus-like particle from PSNE-Pico1. More to the point, it has been described as a particle similar to this one in the microvilli. For the above reasons, it can be asserted that the VLP in the nucleus is not PSNE-Pico1, so even if a VLP of the same form were in the microvilli, the argument that it is a horizontally infected PSNE-Pico1 would not hold. It is important to point out, including point 1 above, that in order to correlate the results of morphological observations with nucleic acid sequence information, sufficient validation is needed to correlate them.

Are there any results of community analysis of phytoplankton samples from the July time period, when PSNE-Pico1 was actively detected? Are there any results from the quantification of viral RNA sequences in phytoplankton pellets, e.g., whether the feeding diatoms were infected by the virus? If many of the feeding diatoms have been infected by the virus, then this would allow for a deeper discussion in this regard.

Reviewer #2

(Remarks to the Author)

This is an interesting study that reports four novel RNA viruses associated with copepods and then examines the potential interactions among the viruses and the copepods. It is quite a comprehensive study, spanning from transcriptomics to identify the viruses and then study viral-host gene expression relationships, an extensive time course of monitoring, environmental community composition work, potential predator quantifications, and copepod genetic structure. The manuscript is well written and clear, and the figures are nicely constructed and reader friendly. I have one major point of contention.

In some of the text (e.g., line 291-310), the authors make a causation conclusion for the pico1 virus and copepod physiology and numbers. However, this is an overstep as the data right now are only correlative. It is interesting that there are VLPs similar to those seen in the gut prey cells, but that is still not enough to conclude it is the same virus. The invertebrate-associated members of the Marnaviridae are all from filter-feeders that are almost certainly consuming large numbers of protists. The veracity of these as invertebrate-infecting viruses has not been demonstrated.

This does not diminish my enthusiasm for the study, but these particular conclusions are not justified and the relevant text throughout the manuscript needs edited.

Andrew Lang

Version 1:

Reviewer comments:

Reviewer #1

(Remarks to the Author)

The authors' revised version and their responses to the reviewers clearly demonstrate that they have interpreted their data and revised their paper in good faith. They have also provided satisfactory responses to the reviewers' critical comments. While there is no reliable evidence of horizontal transmission, I believe there is no issue with the fact that "horizontal transmission" is described with caution, as it remains the central message of this paper.

Additionally, I appreciate the revision made to the description of the image, which initially suggested the presence of VLPs within the microvilli. The revised phrasing, "around the microvilli of copepod intestines" is more accurate since it remains uncertain whether the VLPs are actually located within the microvilli. I also agree with the authors' decision to retain this image in the paper, as it is visually impactful.

The authors suggest that the VLPs increasing in diatom nuclei may be diatom DNA viruses, and they further elaborate on the possibility that these, in addition to RNA viruses, may be involved. I find this to be a meaningful revision. Previously, I noted a contradiction regarding this point because I also believed these VLPs were diatom DNA viruses. I now understand that the revised version offers a solid discussion on this topic.

Furthermore, the newly presented data convincingly demonstrate that PSNE-Pico1 is derived from phytoplankton and is ingested by copepods, thus strengthening the paper's content.

Based on the above, I conclude that the authors have addressed the comments in good faith, and the paper has been significantly improved. Therefore, I recommend it for publication in Communications Biology.

Reviewer #2

(Remarks to the Author)

Thanks for your revisions to the manuscript and congratulations on the nice work.

COMMSBIO-24-2359-T

Manuscript title: Ecological interactions between marine RNA viruses and planktonic copepods

Author's comments:

We thank the editors and reviewers for their thoughtful comments and suggestions, which resulted in improving our manuscript. We have amended the paper in accordance with the reviewers' recommendations. We also formatted our revised manuscript and complied the policies of Communications Biology, including the word limit of the abstract (Line 13-24), inclusion of the section "Statistics and Reproducibility" (Line 549-553), and depositions of source data for major results (Line 556). All changes from the original submission were recorded in the revised version of the manuscript. Below we detail our responses to each of the reviewers' comments.

Response to Reviewer #1:

Summary from reviewer #1: This paper focuses on the interaction between zooplankton and viruses, investigating the ecological role of viruses in the zooplankton copepod *Pseudocalanus newmani*. It analyzes transcriptome data from *P. newmani* and discovers four novel RNA viruses, and further mentions that some of these viruses regulate the population dynamics of *P. newmani*. This is the first study to detail the impact of viruses on marine zooplankton, providing an important perspective on understanding marine ecosystems. However, there are some extremely important issues in this paper that need to be revised.

The authors' discussion on the possibility of horizontal transmission of the viruses detected in this study (PSNE-Pico1 and PSNE-Narna) is not scientific and needs more thorough verification. In their discussion, they mention that viral infection of *P. newmani* has occurred because the virus was detected when *P. newmani* was cultured in filtered seawater (L243-245), but it is unclear how it was cultured. Demonstrating whether the gut contents were completely digested during that incubation period, or showing that an increase in viral genome occurred even after the gut contents were gone, would indicate that virus infection of *P. newmani* had occurred, but such data were not provided. With these insufficient evidences, I believe that PSNE-Pico1, which was particularly numerous, only detected viruses infecting the feeding diatoms, making horizontal transmission doubtful.

Response: We thank the reviewer for pointing out the importance of our study and providing valuable comments. We agree that we couldn't show clear evidence for horizontal transmission of some viruses including PSNE-Pico1 detected in our study. Although we have added new data including results of RT-qPCR for filtered seawaters (Supplementary Figure 4), we have toned down horizontal transmission in the revised version of the manuscript (Line 61-63, 256-258, 263-270, 305-307, 336-339 in the revised version of the manuscript). Even if no clear evidence is provided for

horizontal transmission, we believe that the importance of this study is not diminished as a first step for understanding the ecological roles of viruses associated with marine zooplankton. In fact, there are both propagative and non-propagative plant viruses affecting the physiology of insect vectors in terrestrial ecosystems (Line 333-336). In addition, as pointed out by the reviewer, we have added explanations for the culture condition of *P. newmani* (Line 383-385). We incubated copepods for 6 hours in filtered seawater, which is enough to remove gut contents (Tsuda & Nemoto 1987). However, Frade et al. (2014) reported phytoplankton viruses vectored by copepods tend to be retained in guts beyond the regular gut clearance time of copepods, and these discussions have been added in the revised version of the manuscript (Line 265-267). We have also added discussions on future works for incubation experiments with empty gut to investigate virus infection in copepods (Line 349-350). Please see our detailed response below as well.

References

- Tsuda, A. & Nemoto, T. The effect of food concentration on the gut clearance time of *Pseudocalanus minutus* Krøyer (Calanoida: Copepoda). *J. Exp. Mar. Bio. Ecol.* **107**, 121-130 (1987).
- Frada, M. J. et al. Zooplankton may serve as transmission vectors for viruses infecting algal blooms in the ocean. *Curr. Biol.* **24**, 2592-2597 (2014).

Major issue from reviewer #1 [1]: Virus-like particles have also been observed in *P. newmani* tissue microvilli in the gut tract (Fig. 4b). It is extremely important to verify whether these particles are viruses. It is necessary to measure and verify the size of the VLPs in multiple images to objectively determine if the size of these particles is roughly the same as the virus-like particles found in the cells of the feeds, which are also present in the gut tract. Furthermore, it is not possible to determine from this image whether VLPs are really present in the microvilli. In this study, TEM observations of 80-90 nm thick sections were observed, and when a typical diatom virus is around 30 nm and the virus particles are on the surface of the microvilli, if they are observed in a single section, then VLPs within the VLPs could appear to be present within the microvilli. To observe such a small VLP by TEM and confirm its presence within the microvilli, it would need to be observed three-dimensionally, e.g. using tomography.

Response: We agree with the reviewer that it's important to measure the size of VLPs. Hence, in the revised version of the manuscript, we have added the analysis of size measurements of VPLs in the Results (Line 186-187) and the Methods (Line 473-474). The size was almost same as a typical diatom virus, and there was no significant difference in diameters of VLPs between gut contents and copepods. In addition, we also agree that it's difficult to show if VLPs are same between the gut contents and copepods, as discussed in the revised version of the manuscript (Line 267-269), and we have added the description that other approaches such as fluorescence *in situ* hybridization can be

useful for studying detection of a specific virus and virus localization (Line 357-359). Additionally, we also agree with the reviewer that it's difficult to show VLPs within the microvilli of copepods. We have toned down the presence of VLPs in the microvilli in the revised version of the manuscript, and we use the word 'around microvilli' and not 'in microvilli' (Line 184-185, Line 765). Regardless of the difficulty in showing the presence of VLPs within microvilli, we believe the results of VLPs observed around copepod microvilli are valuable, and we keep Fig. 4b in our manuscript.

Major issue from reviewer #1 [2]: As mentioned above, it may be difficult to consider the results shown alone from the perspective of horizontal transmission of *P. newmani* by a virus that infects the feed. To show that the virus is infecting *P. newmani*, it is important to confirm viral amplification in the cells and show data on physiological increases from the initial viral load. The authors mention that there are no data directly showing these (L314-415), so discussions based on horizontal transmission should be avoided as much as possible. On the other hand, it is possible, for example, that the feeding plankton may have indirectly affected *P. newmani* physiology by decreasing its feeding quality as a result of the viral infection. In addition to horizontal transmission, these multifaceted possibilities should be equally mentioned.

Response: We thank the reviewer for the constructive comments. As mentioned in the response to "Summary from reviewer #1", we avoided horizontal transmission of viruses to copepods in the revised version of the manuscript. We have added the discussion on a future work to confirm viral amplification in copepods in the revised version of the manuscript (Line 349-350). In addition, we agree that infected phytoplankton could affect *P. newmani* physiology, and this is an important point to be discussed. We have added indirect effects on copepod physiology by decreases in food quality due to viral infections on food sources in the revised version of the manuscript (Line 340-342). To our knowledge, no studies focused on the physiological changes of copepods feeding on infected phytoplankton, and we believe that this kind of discussion would provide a window into the role of the virus in marine ecosystems.

Major issue from reviewer #1 [3]: The authors state that VLPs of gut tract contents are observed in the nuclei of phytoplankton cell-like particles (Fig. 4a, L167-168); as this sample is a *P. newmani* sample from a time when PSNE-Pico1 was detected in large numbers in M&M, the authors suggest that the VLPs in the gut contents are PSNE-Pico1 particles. On the other hand, diatom RNA viruses multiply in the cytoplasm and not in the nucleus, so if this VLP is in the nucleus, it is a different virus-like particle from PSNE-Pico1. More to the point, it has been described as a particle similar to this one in the microvilli. For the above reasons, it can be asserted that the VLP in the nucleus is not PSNE-Pico1, so even if a VLP of the same form were in the microvilli, the argument that it is a

horizontally infected PSNE-Pico1 would not hold. It is important to point out, including point 1 above, that in order to correlate the results of morphological observations with nucleic acid sequence information, sufficient validation is needed to correlate them.

Response: We appreciate the comment from the reviewer. According to the reviewer's comments, we have thoroughly reviewed papers associated with algae viruses. As mentioned by the reviewer, diatom RNA viruses multiply not in the nucleus but in the cytoplasm (Lang et al. 2021). On the other hand, there are ssDNA viruses of diatom which multiply in the diatom nucleus (Nagasaki et al. 2005). Thus, we have added the discussion that VLPs observed by TEM analysis might be different from PSNE-Pico1, indicating more viruses associated with copepods (Line 277-279). As mentioned in other responses to the reviewer's comments, horizontal transmission is avoided in the revised version of the manuscript, and we have added size information of VLPs (Line 186-187). Although same diameters of VLPs were observed between gut contents and copepods, we have the discussion that it's difficult to conclude if VLPs are same viruses between gut contents and copepods only based on the TEM analysis (Line 267-269). We believe that TEM results are still valuable, even if viruses detected from gut contents are not PSNE-Pico1, since this is a good result that phytoplankton viruses are detected in the copepod gut when there are large physiological changes based on transcriptome analysis. Thus, we didn't remove results of the TEM analysis in the revised version of the manuscript.

References

- Lang, A. S. et al. ICTV Virus Taxonomy Profile: Marnaviridae 2021. *J. Gen. Virol.* **102**, 001633 (2021).
- Nagasaki, K. et al. Previously unknown virus infects marine diatom. *Appl Environ Microbiol.* **71**, 3528-3535 (2005).

Major issue from reviewer #1 [4]: Are there any results of community analysis of phytoplankton samples from the July time period, when PSNE-Pico1 was actively detected? Are there any results from the quantification of viral RNA sequences in phytoplankton pellets, e.g., whether the feeding diatoms were infected by the virus? If many of the feeding diatoms have been infected by the virus, then this would allow for a deeper discussion in this regard.

Response: We thank the reviewer for the comments. We have the results of community analysis of Eukaryota during the sampling period in Figure 5. In addition, we have added OTU-level community analysis for both gut contents and ambient water samples using 18S metabarcoding (Supplementary Figure 2). As mentioned in the manuscript, food sources were not largely changed when PSNE-Pico1 was actively detected in OTU-level analysis. As for results of RNA viruses in phytoplankton cells,

there were no results provided in the original submission. Fortunately, two filter samples collected in March and July were available. Although comprehensive RNA-seq data are not available, we have carried out RT-qPCR to detect four RNA viruses in our study. As a result, PSNE-Pico1 was only detected in July but not in March (Supplementary Figure 4; Line 261-262 in the revised version of the manuscript), and other viruses were not detected in both months. This is an important result to support that PSNE-Pico1 is originated from phytoplankton and ingested by copepods, and we thank the reviewer for the valuable advice.

Supplementary Figure 2 in the revised version of the manuscript.

Supplementary Figure 4 in the revised version of the manuscript.

Response to Reviewer #2 (Andrew Lang):

Summary from reviewer #2:

This is an interesting study that reports four novel RNA viruses associated with copepods and then examines the potential interactions among the viruses and the copepods. It is quite a comprehensive study, spanning from transcriptomics to identify the viruses and then study viral-host gene expression relationships, an extensive time course of monitoring, environmental community composition work, potential predator quantifications, and copepod genetic structure. The manuscript is well written and clear, and the figures are nicely constructed and reader friendly. I have one major point of contention.

In some of the text (e.g., line 291-310), the authors make a causation conclusion for the Pico1 virus and copepod physiology and numbers. However, this is an overstep as the data right now are only correlative. It is interesting that there are VLPs similar to those seen in the gut prey cells, but that is still not enough to conclude it is the same virus. The invertebrate-associated members of the *Marnaviridae* are all from filter-feeders that are almost certainly consuming large numbers of protists. The veracity of these as invertebrate-infecting viruses has not been demonstrated. This does not diminish my enthusiasm for the study, but these particular conclusions are not justified and the relevant text throughout the manuscript needs edited.

Response: We are grateful for the positive comments from the reviewer. We agree with the reviewer's concern that we need a causal conclusion for the Pico1 virus, because we didn't provide direct evidence of horizontal transmission of Pico1 to copepods. This issue was also pointed out by reviewer 1, and we have corrected the manuscript according to the reviewers' suggestions. We have avoided having discussions on horizontal transmissions through the manuscript, and indirect effects without horizontal transmission was also discussed, because there might be decreased food quality by viral infections of phytoplankton (Line 340-342). In addition, we have added the discussion that it's difficult to conclude if VLPs are same viruses between gut contents and copepods only based on the TEM analysis (Line 267-269). As I respond to reviewer 1, even if no clear evidence is provided for horizontal transmission, we believe this is the first study to detail the impact of viruses on marine zooplankton, and the results obtained in this study are valuable as a first step to understand ecological roles of viruses associated with marine zooplankton. All changes are recorded in the revised version of the manuscript, and please see the responses to reviewer 1. We have carefully corrected our manuscript, especially for the results of Pico1.

COMMSBIO-24-2359-A

Manuscript title: Ecological interactions between marine RNA viruses and planktonic copepods

Author's comments:

We thank the editors and reviewers for their comments. Because both reviewers did not request further revision, we have corrected our manuscript only based on editorial requests.

Response to Reviewer #1:

Summary from reviewer #1: The authors' revised version and their responses to the reviewers clearly demonstrate that they have interpreted their data and revised their paper in good faith. They have also provided satisfactory responses to the reviewers' critical comments. While there is no reliable evidence of horizontal transmission, I believe there is no issue with the fact that "horizontal transmission" is described with caution, as it remains the central message of this paper. Additionally, I appreciate the revision made to the description of the image, which initially suggested the presence of VLPs within the microvilli. The revised phrasing, "around the microvilli of copepod intestines" is more accurate since it remains uncertain whether the VLPs are actually located within the microvilli. I also agree with the authors' decision to retain this image in the paper, as it is visually impactful.

The authors suggest that the VLPs increasing in diatom nuclei may be diatom DNA viruses, and they further elaborate on the possibility that these, in addition to RNA viruses, may be involved. I find this to be a meaningful revision. Previously, I noted a contradiction regarding this point because I also believed these VLPs were diatom DNA viruses. I now understand that the revised version offers a solid discussion on this topic. Furthermore, the newly presented data convincingly demonstrate that PSNE-Pico1 is derived from phytoplankton and is ingested by copepods, thus strengthening the paper's content. Based on the above, I conclude that the authors have addressed the comments in good faith, and the paper has been significantly improved. Therefore, I recommend it for publication in Communications Biology.

Response: We thank the reviewer for recommending our manuscript for publication in Communications Biology without an additional revision. We have corrected our manuscript only based on editorial requests.

Response to Reviewer #2:

Summary from reviewer #2:

Thanks for your revisions to the manuscript and congratulations on the nice work.

Response: We thank the reviewer for recommending our manuscript for publication in Communications Biology without an additional revision. We have corrected our manuscript only based on editorial requests.